# Evaluation of Postoperative Pain Following Single-Visit Root Canal Treatment with Rotary and Reciprocal Ni–Ti File Systems in Children

**DOI:** 10.3390/medicina58010050

**Published:** 2021-12-29

**Authors:** Alp Abidin Atesci, Aslı Topaloglu-Ak, Ece Turan, Ozant Oncag, Mehmet Emin Kaval

**Affiliations:** 1Independent Researcher, Izmir 35220, Turkey; Alpabidinatesci91@gmail.com (A.A.A.); ece.turan@ege.edu.tr (E.T.); 2Department of Pediatric Dentistry, School of Dentistry, Istanbul Aydın University, Istanbul 34295, Turkey; 3Department of Pediatric Dentistry, School of Dentistry, Ege University, Izmir 35040, Turkey; ozant.oncag@ege.edu.tr; 4Department of Endodontics, School of Dentistry, Ege University, Izmir 35040, Turkey; Mehmet.emin.kaval@ege.edu.tr

**Keywords:** postoperative pain, root canal treatment, reciprocating motion, rotary motion

## Abstract

*Background and Objectives*: Postoperative pain is a common symptom of a flare-up after root canal treatments (RCTs). Insufficient instrumentation, extrusion of irrigation solutions and debris, and the existence of a periapical lesion are the factors affecting postoperative pain after root canal treatments. The aim of this study was to evaluate the postoperative pain and instrumentation time of the single-file reciprocating system and multiple-file Ni–Ti rotary system in children ages 9–12 years old. *Materials and Methods*: Our study was conducted on 51 permanent mandibular molars with the diagnosis of irreversible pulpitis. Patients were randomly assigned into two groups, and RCTs were completed with either the Reciproc Blue or Protaper NEXT file systems. Instrumentation time for each system was noted, and patients were given a pain scale that included a visual analog scale for 6, 24, 48, and 72 h after treatment. Postoperative pain scores and instrumentation times were analyzed statistically with a chi-square test and Student’s *t*-test. *Results:* There was no statistically significant difference in postoperative pain between the Reciproc Blue and Protaper NEXT systems at all time intervals. Instrumentation time was significantly shorter in the Reciproc Blue group in comparison with the Protaper NEXT group. *Conclusions:* Postoperative pain findings following RCT using single-file reciprocating systems were similar to the rotary system group. However, chair time in the reciprocating system group was significantly lower. This provided a comfortable and patient-friendly treatment approach for children, and could enhance their cooperation.

## 1. Introduction

Postoperative pain is one of the most common complications of root canal treatment, and according to a systematic review, the prevalence rate is reported to be between 3–58% [1]. In addition to this, it has been reported that the postoperative pain intensity was observed highest within the first 6 h and decreased over time [2]. Several risk factors are associated with postoperative endodontic pain, such as; age, gender, extrusion of debris, tooth anatomy, residual pulp remnants, preoperative pain, use of analgesic agents, pulpal and periapical status, and several treatment visits [3,4,5]. Adequate disinfection of the root canal system is crucial to minimize the symptoms, such as pain, abscess, and cellulite. For this purpose, root canal instrumentation and irrigation procedures should be performed efficiently; additionally, lasers and ultrasonic devices can be considered as adjunctive protocols [6].

The apical extrusion of debris during instrumentation is considered the primary cause of postoperative endodontic pain [7]. In addition, the quantity of debris extrusion has been linked with the mechanical shaping procedures [4]. Several studies reported that nickel–titanium (Ni–Ti) rotary shaping extrude less debris and cause less postoperative pain compared with the stainless steel hand files [8]. The rotary action of Ni–Ti instruments combined with copious irrigation reduces the risk of postoperative pain [9].

The use of single-file reciprocating instruments has several advantages, such as reduced number of instruments, treatment time, cost, and elimination of cross-contamination [10]. A recent study reported that both Ni–Ti rotary and reciprocating instruments were efficient in reducing the endotoxins and bacterial byproducts in infected root canals.

The comparison of Ni–Ti rotary instruments with reciprocating instruments is controversial [11,12]. A continuous rotary motion may improve the coronal movement of infected debris and result in less debris extrusion to the periapical region than the reciprocal motion [11]. On the other hand, Neelakantan et al. [13] and Deus et al. [12] reported that Ni–Ti rotary instruments showed significantly higher apical debris transportation compared to reciprocal instruments, leading to increased postoperative pain. A recent meta-analysis of randomized clinical trials showed that rotary instruments were associated with a lower prevalence of postoperative pain than reciprocating instruments [14].

The success rates of single- and multiple-visit endodontic treatment were found to be similar, and according to several studies, patients tolerate and prefer single-visit treatment because of the reduced treatment time and operative procedures [2,15,16,17]. Less chair time in pediatric patients is also essential. To this date, there were no studies found in the literature that compare postoperative pain and instrumentation time of single-visit root canal treatment done with single-file reciprocating systems and multiple-file rotary systems in permanent teeth of children. Therefore, the aim of this randomized clinical study was to evaluate the postoperative pain and instrumentation time of the single-file reciprocating system and multiple-file Ni–Ti rotary system in children ages 9–12.

## 2. Material and Methods

All clinical procedures in the present study were approved by the Ethics Committee on Human Research, Ege University, Izmir, Turkey (20-7T/55) and registered at www.clinicaltrials.gov (NCT04510571), accessed on 9 January 2019. The sample size was calculated according to an alpha error level of 0.05 and a power of 0.8 ^10^, with an estimated dropout of 20%, requiring 25 teeth for each group. The present study enrolled 50 children aged 9–12 who were referred to the Department of Pedodontics, Ege University, and required RCT of a mature first mandibular molar. Cooperative patients with no systematic disorders and allergic reaction to NaOCl who had their first permanent mandibular molar teeth diagnosed with symptomatic irreversible pulpitis with no periapical pathology or abscess were included in the study. Parents were given information about the study design, and informed consent was obtained from every patient’s parents. Exclusion criteria were as follows: patients who were on antibiotics or analgesics preoperatively.

The patients were examined by another pediatric dentist who was not aware of the study design, and the randomization was performed as two equal groups by using an online randomization software (http://www.random.org, accessed on 9 January 2019) [18]. Patients were allocated sequential numbers in the order of enrollment following the randomization process, and the operator was informed regarding the treatment modality, either with Protaper NEXT or with Reciproc Blue files.

Two experienced pediatric dentists performed a single-visit root canal treatment. Before treatment, a standardized periapical radiograph was taken for each tooth. Local anesthesia with 2% lidocaine hydrochloride and 1:100,000 adrenaline (JETOKAIN, Adeka İlaç San. ve Tic. A.Ş., Istanbul, Turkey) was administered. After local anesthesia, isolation of teeth was done with rubber dam application. Caries removal and access cavity preparation was performed with round diamond burs under constant water cooling. Working lengths were measured with stainless steel K files (Dentsply, Maillefer, Tulsa, OK, USA) using an electronic apex locator (Raypex 5, VDW, München, Germany), and a glide path was established using stainless steel K files.

## 3. Reciproc Blue (VDW, Munich, Germany) Group (*n* = 25)

The canals were shaped in accordance with the manufacturer’s recommendations. An R25 (25.08) instrument was introduced into the root canal with slow pecking movements within a 2 mm range each time. The flutes and remnants were cleaned after 3 pecking moves. Then, an R40 instrument (40.06) was selected to shape the distal canals as the #20 K file was passively introduced to the working length. The root canals were irrigated with 2 mL of 2.5% sodium hypochlorite after each instrument change.

## 4. Protaper Next (Dentsply Maillefer, Ballaigues, Switzerland) Group (*n* = 25)

The instrumentation was performed according to manufacturer’s recommendations. X1 and X2 (25.06) files were used to shape mesial and distal canals, and then X3 and X4 (40.06) instruments were used to enlarge only distal root canals. The root canals were irrigated with 2 mL of 2.5% sodium hypochlorite after each instrument change.

The amount of irrigation was equal for each tooth; as for the final irrigation to remove the smear layer, root canals were irrigated with 17% EDTA. All root canals were dried with sterile paper points and obturated with compatible gutta-percha master cones and AH Plus sealer (Dentsply Maillefer, Ballaigues, Switzerland). Obturation of the root canals was performed by a single-cone cold lateral condensation technique. Following obturation, glass ionomer cement (Ketac Molar Easy, 3M ESPE, St. Paul, MN, USA) was used for temporary restoration. A postoperative periapical radiograph was taken for each tooth to evaluate the quality of the obturation. The time for root canal instrumentation was recorded after glide path establishment until the end of instrumentation, including total active instrumentation, cleaning of the flutes of the instruments, and irrigation between file changes within the sequence.

Postoperative pain scores were scored using a modified Wong–Baker pain rating scale [19,20]. The patients and parents were given detailed information, and were asked to select the pain responses from the pain rating scale (0 = no pain, 1 = slight pain, 2 = moderate pain, 3 = severe pain) at 6 h, 24 h, 48 h, and 72 h postoperatively. The statistical analysis of the results was done using IBM SPSS Statistics 25.0 (SPSS Inc., Chicago, IL, USA) using the chi-square test and *t*-test. Statistical significance was defined as *p* < 0.05.

## 5. Results

### 5.1. Demographic Data

Of 50 patients, 25 were males and 25 were females; the mean age of the patients was 10.9 years in the Reciproc Blue group and 11.1 in the Protaper NEXT group.

### 5.2. Postoperative Pain Evaluation

Results showed that there were no significant differences between genders and operators in terms of postoperative pain. The mean postoperative pain scores for different time intervals are shown in Table 1. Statistical analysis revealed no significant differences regarding postoperative pain between the Reciproc Blue and Protaper NEXT file systems at all time intervals (*p* > 0.05). Postoperative pain was highest at 6 h in both groups, and decreased over time.

### 5.3. Instrumentation Time Evaluation

The mean time taken for instrumentation of the root canals is shown in Table 2. Instrumentation time of multiple-file the Protaper NEXT rotary system was significantly longer than that of the single-file Reciproc Blue system (*p* < 0.05).

## 6. Discussion

Managing postoperative pain is essential for endodontic treatment, and determines clinical success [21]. Therefore, the primary aim of this study was to investigate the effects of Protaper NEXT versus Reciproc Blue on the intensity of postoperative pain after root canal treatment in children. In order to make an accurate comparison and increase the validity of the study, variables such as the type and quantity of irrigation were kept similar. According to the results of this study, postoperative pain after root canal treatment with the multiple-file rotary system and the single-file reciprocating system were similar. In addition, the instrumentation time with a single-file reciprocating system was significantly shorter than with a multiple-file rotary system.

Postoperative pain is associated with various factors, such as final apical size, overinstrumentation, chemical irritants, incomplete removal of the pulp tissue, hyperocclusion, and extrusion of debris into the periapical region [7,20]. Most studies in the literature based on postoperative pain investigated the amount of apically extruded debris, as it is one of the primary causes of pain. Extrusion of debris and microorganisms into the periapical area causes inflammation, leading to pain [10]. In addition, it is reported that debris containing bacteria and their byproducts may be packed laterally in isthmuses and protrusions during root canal preparation, and could reinfect the root canal system or cause postoperative pain [22]. Continuous forward movement of the rotary files enables debris transportation coronally, while forward and backward motion of reciprocating files causes debris to be packed in protrusions and isthmus areas [23].

A large number of studies investigated the effect of various instrumentation systems on postoperative pain. According to a recent meta-analysis by Sun et al. [24] and Caviedes-Bucheli et al. [25], the use of multiple-file rotary systems showed a lower incidence of postoperative pain than the single-file reciprocating systems. In addition, several studies also found lower postoperative pain intensity for multiple-file rotary systems in comparison to single-file reciprocating systems [26]. On the contrary, Neelakantan & Sharma ^13^ and Shokraneh [27] reported higher pain intensity with multiple-file rotary systems when compared to single-file reciprocating systems. These inconsistencies might be explained by the differences in the methodologies of these studies, such as different types and concentrations of irrigants used in these studies. There was no significant difference between rotary and reciprocating file systems regarding postoperative pain in the present study. The results of this study were in accordance with the previous studies [15,28]. However, to date, there has been no data published in terms of comparing postoperative pain between multiple-file rotary and single-file reciprocating systems among children.

Single-visit root canal treatment has several advantages over multiple-visit root canal treatment, such as reductions in total treatment time, complications due to repeated injections, and microleakage of temporary restorations. Previous studies reported no significant difference in postoperative pain between single-visit and multiple-visit root canal treatment [4,29]. Although the success rate and postoperative pain prevalence were similar between single- and multiple-visit root canal treatment, patients preferred single-visit treatment due to the reduced treatment duration [15,16].

In the present study, the Reciproc Blue system required a significantly shorter time for instrumentation because only one file was used. On the other hand, four files were used in the Protaper NEXT system. The results of the present study were in accordance with the previous studies [10,30]. It was reported that in children, increased postoperative pain was associated with anxiety during treatment [31]. Considering the fact that the Reciproc Blue file system requires significantly shorter root canal preparation time, its application may be more appropriate for children in order to work faster and manage behavior management problems, such as anxiety.

## 7. Conclusions

Postoperative pain was found to be similar in both the Reciproc Blue and Protaper NEXT file systems at 6 h, 24 h, 48 h, and 72 h intervals. However, the highest pain intensity was observed at 6 h intervals, and decreased significantly over time. In addition, root canal preparation time with the single-file Reciproc Blue system was significantly shorter compared to multiple-file Protaper NEXT system. Therefore, root canal treatment with the single-file Reciproc Blue system is an appropriate choice for RCT for pediatric patients due to the reduced instrumentation time.

## 8. Points of Clinical Significance

It is essential to reduce postoperative pain in children to reduce anxiety, and there were no significant differences observed between the single-file reciprocating systems and multiple-file rotary systems.Treatment time, especially in children, is an essential parameter for behavior management, and root canal instrumentation with single-file reciprocating systems requires significantly less instrumentation time compared to the multiple-file rotary systems.There were no studies found in the literature that compared postoperative pain and instrumentation time of root canal treatment done with single-file reciprocating systems and multiple-file rotary systems in children’s permanent teeth.

## Figures and Tables

**Table 1 medicina-58-00050-t001:** Mean postoperative pain scores for Reciproc Blue and Protaper NEXT file systems at different time intervals.

Hours	Reciproc Blue	Protaper NEXT	*p*-Value
6 h	0.85	1.05	0.526
24 h	0.10	0.20	0.971
48 h	0.10	0.15	0.574
72 h	0	0.15	0.152

**Table 2 medicina-58-00050-t002:** Mean preparation time (s) for Reciproc Blue and Protaper NEXT file systems.

	Meantime (s)	Standard Deviation
Reciproc Blue	430	58
Protaper NEXT	699	65

## Data Availability

All the data are available from the corresponding author upon reasonable request.

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
