# Peer review of "Evaluation of Postoperative Pain Following Single-Visit Root Canal Treatment with Rotary and Reciprocal Ni–Ti File Systems in Children"

_medicina, 2021, doi:10.3390/medicina58010050_

Round 1
Reviewer 1 Report
Methods:
- How were the patients recruited? What was the sample population?
- Were the patients of two groups aware of the different in treatment procedures they received?
Results:
- Is the age rage of the patients available?
- What were the range of level of post operative pain? The difference helps understand the effect of the two approaches.
- There were 50 patients. In the abstract, it stated that 51 permanent molars were included in the study. Was some patient had more than one tooth treated?
- Why difference between 2 operators in terms of post-operative pain was mentioned? Were the patient treated by a single experienced paediatric dentists only?
General:
- There are a few typos and formatting issues.
Author Response
Methods:
- How were the patients recruited? What was the sample population?
The present study enrolled fifty children aged between 9-12 who were referred to the Department of Pedodontics, Ege University and required RCT of a non-vital asymptomatic mature first mandibular molar. Inclusion criteria were as follows: co-operative patients with no systematic disorders and allergic reaction to NaOCl. Parents were given information about the study design and informed consent was obtained from every patients' parents. First permanent mandibular molar teeth which had been diagnosed with symptomatic irreversible pulpitis with no periapical pathology or abscess were included in the study. Exclusion criteria were as follows: patients who were on antibiotics or analgesics preoperatively. (this information is added in the text)
- Were the patients of two groups aware of the different in treatment procedures they received?
Parents were given information about the study design and informed consent was obtained from every patients' parents. However they did not know which procedure was applied to them in order to not bias the results.
Results:
- Is the age rage of the patients available?
The age range of the patients varied from 9 to 12. (This information is given in the Material methods section)
- What were the range of level of post operative pain? The difference helps understand the effect of the two approaches.
??
- There were 50 patients. In the abstract, it stated that 51 permanent molars were included in the study. Was some patient had more than one tooth treated?
Yes, one patient had two molars treated, that is why we ended up with 51 first permanent molars root canal filled.
- Why difference between 2 operators in terms of post-operative pain was mentioned? Were the patient treated by a single experienced paediatric dentists only?
One experienced pediatric dentist completed all the treatments.
General:
- There are a few typos and formatting issues.

Reviewer 2 Report
The introduction has to provide more information.
I recomand the authors to include also a chapter about the root disinfection with laser because this is a very helfull tool in endodontics.
They could get information from this article: DOI 10.1016/j.pdpdt.2019.101611
The authors have to describe better what were the inclusion and exclusion criteria for the patients included in the study.
The results have to be restructured.
The discussion has to include more recent published articles and has to compare better the results obtained by the study.
The visual analysis scale is subjective.
The conclusion has to be more precise.
Author Response
The introduction has to provide more information.
I recomand the authors to include also a chapter about the root disinfection with laser because this is a very helfull tool in endodontics.
They could get information from this article: DOI 10.1016/j.pdpdt.2019.101611
We sincerely appreciate reviewer’s suggestion the reference is added in the text.
The authors have to describe better what were the inclusion and exclusion criteria for the patients included in the study.
The present study enrolled fifty children aged between 9-12 who were referred to the Department of Pedodontics, Ege University and required RCT of a mature first mandibular molar. Inclusion criteria were as follows: co-operative patients with no systematic disorders and allergic reaction to NaOCl. Parents were given information about the study design and informed consent was obtained from every patients' parents. First permanent mandibular molar teeth which had been diagnosed with symptomatic irreversible pulpitis with no periapical pathology or abscess were included in the study. Exclusion criteria were as follows: patients who were on antibiotics or analgesics preoperatively. (this information is added in the text)
The results have to be restructured.
According to the suggestions of the reviewer results section is divided into sections.
The discussion has to include more recent published articles and has to compare better the results obtained by the study.
2021, lately published reference is added .
The visual analysis scale is subjective.
Patients were asked to scale their pain after the RCT in 6h, 24h, 48h and 72h. Both represented by faces and numbers. These scales are easy to understand for pediatric patients. They are frequently used to to evaluate the post operative pain as they are practical in clinic use.
The conclusion has to be more precise.
In conclusion, two techniques were same in regards to post operative pain after RCT. However, Reciprocal system took less time which is considered to be more appropriate for use in pediatric patients. Less time in the dental chair is both preferred by pediatric patients and pedodontist in order to not to lose the co-operation during the treatment.

Round 2
Reviewer 2 Report
The quality of the article was improved. Congratulations to the team of authors.